# Estimating Double Cropping Plantations in the Brazilian Cerrado through PlanetScope Monthly Mosaics

**Edson Eyji Sano** [1,*], **Édson Luis Bolfe** [2,3], **Taya Cristo Parreiras** [3], **Giovana Maranhão Bettiol** [1], **Luiz Eduardo Vicente** [4], **Ieda Del'Arco Sanches** [5] and **Daniel de Castro Victoria** [2]

1   Brazilian Agricultural Research Corporation, Embrapa Cerrados, Planaltina 73301-970, DF, Brazil
2   Brazilian Agricultural Research Corporation, Embrapa Agricultura Digital, Campinas 13083-886, SP, Brazil
3   Institute of Geosciences, State University of Campinas (UNICAMP), Campinas 13083-855, SP, Brazil
4   Brazilian Agricultural Research Corporation, Embrapa Meio Ambiente, Jaguariúna 13917-200, SP, Brazil
5   National Institute for Space Research (INPE), São José dos Campos 12227-010, SP, Brazil
*   Correspondence: edson.sano@embrapa.br

**Abstract:** Farmers in the Brazilian Cerrado are increasing grain production by cultivating second crops during the same crop growing season. The release of PlanetScope (PS) satellite images represents an innovative opportunity to monitor double cropping production. In this study, we analyzed the potential of six PS monthly mosaics from the 2021/2022 crop growing season to discriminate double cropping areas in the municipality of Goiatuba, Goiás State, Brazil. The four multispectral bands of the PS images were converted into normalized difference vegetation index (NDVI), enhanced vegetation index (EVI), green–red normalized difference index (GRNDI), and textural features derived from the gray-level co-occurrence matrix (GLCM). The ten most important combinations of these attributes were used to map double cropping systems and other land use and land cover classes (cultivated pasture, sugarcane, and native vegetation) of the municipality through the Random Forest classifier. Training and validation samples were obtained from field campaigns conducted in October 2021 and April 2022. PS mosaic from February 2022 was the most relevant data. The overall accuracy and Kappa index of the final map were 92.2% and 0.892, respectively, with an accuracy confidence of 81%. This approach can be expanded for mapping and monitoring other agricultural frontiers in the Cerrado biome.

**Keywords:** Random Forest; gray-level co-occurrence matrix; tropical savanna; land use and land cover mapping; satellite constellation

## 1. Introduction

The Brazilian tropical savanna (Cerrado) covers an area surpassing 200 million ha in the central part of the country and is one of the richest biomes in the world in terms of biodiversity, includes more than 12,400 plant species that are endemic and has unique adaptation to the occurrences of fire, soils with low fertility and high acidity, and a six-month dry season [1]. It is considered as one of the world´s hotspots for biodiversity conservation [2]. Roughly, half of this biome is covered by native vegetation [3], which corresponds to a mosaic of different species of grasses, shrubs, and trees in different proportions, depending on the soil texture, soil structure, soil moisture, and rainfall conditions, among other factors [4]. The Cerrado is also well-known for its highly mechanized large-scale production of food and energy [5]. In this biome, we find 47 million ha of cultivated pastures, mostly the African Brachiaria species, 23 million ha of annual crops, mainly soybean, maize, and cotton, and 500 thousand ha of perennial and semi-perennial crops, mostly coffee and sugarcane [6]. Pasturelands are spread throughout the entire biome while croplands are cultivated in more specific regions, that is, in agricultural frontiers with extensive flat terrains and rainfall levels typically higher than 1000 mm per year in the rainy season (from October to March) [1].

According to the Brazilian Native Vegetation Protection Law no. 12,651 of 25 May 2012, also known as the Brazilian Forest Code, farmers located in the Brazilian Cerrado must keep at least 20% of their rural properties with native vegetation or at least 35% if the property is located in the Amazon-Cerrado ecotone (Brazilian Legal Amazon) [7]. Furthermore, only 7.2% of the Cerrado is protected in terms of conservation units [1]. This soft environmental regulation reinforces the fact that both national and international conservation efforts tend to prioritize rainforests and to consider savanna ecosystems as land reserves for agricultural expansion [8,9]. For example, the Amazon Soy Moratorium, proposed to reduce soybean expansion even in the Brazilian Amazon, effectively reduced the conversion of primary or secondary forests into soybean plantations in this region [10]. However, it caused a side effect in the Brazilian Cerrado, encouraging the opening of new agricultural areas, mainly in the new agricultural frontier known as MATOPIBA [11,12].

At the same time, farmers in the Brazilian Cerrado are adopting different land management techniques to improve profitability and to reduce climatic risks (e.g., dry spell occurrences and low rainfall conditions in some years). Among different techniques, we can point out the use of the double cropping management system, which is becoming quite popular mainly in the consolidated agricultural frontiers in the Brazilian Cerrado [13,14]. The double cropping system refers to crops that are planted twice in the same area and in the same crop growing season and has been adopted extensively not only in Brazil but also in other countries, especially China [15]. The harvested area of double-cropped maize in Brazil is now about four times higher than that from the single crop system: 16.4 million ha against 4.3 million ha in the 2021/2022 crop season, respectively [16].

Farms with the double cropping system can be identified accurately using in situ surveys. However, they are time consuming, labor intensive, and costly. Several authors have proposed different satellite data and procedures to monitor agricultural fields with double cropping systems since crop intensification is an essential information for land management and trade decisions. Mingwei et al. [17] used an 8-day time series of Moderate Resolution Imaging Spectroradiometer (MODIS) data sets processed by the Fast Fourier Transform to discriminate double cropping systems in Northern China, mainly winter wheat-maize and winter wheat-cotton plantations. Picoli et al. [18] and Chaves et al. [19], among others, proposed the use of the Time-Weighted Dynamic Time Warping (TWDTW) method applied to the MODIS data to identify large-scale crop successions and rotations in an ecotone region between Cerrado and Amazon in the Mato Grosso State, Brazil (soybean as the first crop and maize and millet as the second crop), with promising results. The rationale is the use of logistic functions to find the number of peaks in spectral indices in a specific crop calendar. However, differences in climate, crop type, and planting dates can cause differences in crop growth patterns, affecting the accuracy significantly. Another drawback is the 250 m coarse resolution of the MODIS sensor, which is quite sensitive to the occurrences of multiple targets within a single pixel.

More recently, NASA proposed the Harmonized Landsat and Sentinel-2 (HLS) initiative to produce analysis-ready, surface reflectance images acquired by the Landsat Operational Land Imager (OLI) and Sentinel-2 Multispectral Instrument (MSI) sensors at a 2–4 day frequency and a 10–30 m spatial resolution [20]. Therefore, in theory, these images are more suitable for mapping crop intensification besides presenting the possibility of mapping smaller areas (typically, >2–3 ha), as demonstrated by the studies conducted, for example, by Liu et al. [21] and Pan et al. [22]. Parreiras et al. [23] also demonstrated the potential of HLS to monitor specific crop (soybean) in the western Bahia State, Brazil. However, monitoring the double cropping system in the Cerrado biome using optical remote sensing data is not an easy task, mainly because of the persistent cloud coverage during the rainy season, even if the frequency of data acquisition is reduced to 2–4 days.

With the launch of the PlanetScope (PS) constellation of nanosatellites, we have the opportunity to use the near daily-based satellite data with a pixel size of ~5 m to monitor rainfed crop production with a reasonable number of cloud-free images during the crop growing season. The PS satellite constellation that was launched by the Planet™

is composed of more than 200 CubeSats that acquire daily images in four multispectral bands in the blue, green, red, and near-infrared (NIR) spectral regions [24]. The monthly mosaics of PS, released by Norway´s International Climate & Forests Initiative (NICFI) [25], correspond to the combination of the best daily acquisitions during the month so that they are mostly cloud-free. Consequently, the NICFI images provide the best available multispectral data set to map double cropping systems in tropical regions. Several studies have already demonstrated the potential of PS monthly mosaics for different applications, including tree cover monitoring over tropical regions, based on the U-net deep learning model [26], cropland mapping in smallholder landscapes in Mozambique, based on the Random Forest (RF) machine learning model [27], planted Eucalyptus mortality in Mato Grosso State, Brazil, the logistic regression model [28], and land-use following deforestation in Ethiopia based on the U-net model [29], among others.

However, as pointed out by Wagner et al. [26], the main drawback of NICFI mosaics is the variation in the reflectance values between different Planet satellite sensors and between dates within the same mosaics. As reported in the NICFI documentation [25], the absolute radiometric calibration is not guaranteed in the normalized surface reflectance base maps. Therefore, it seems that the performance of PS monthly mosaics for different applications is site-specific.

In this study, we assessed the potential of PS monthly mosaics acquired from October 2021 to April 2022 (rainy season) to identify double cropping systems and other land use and land cover (LULC) classes in the municipality of Goiatuba, southwestern Goiás State, Brazil. We selected this study area because it is representative of the Cerrado´s municipalities presenting large areas with double cropping systems and because of its regional socio-economic importance. To the best of our knowledge, there is no study evaluating this high spatial and temporal resolution and NICFI´s monthly products to discriminate double crop agricultural fields over the Cerrado biome. Our ultimate goal is to analyze the feasibility of using PS monthly mosaics in an operational way to monitor agricultural areas of the Brazilian Cerrado with double cropping systems. Within this context, this work presents the following scientific contributions: (i) it is the first proposal of mapping the double cropping system in the Brazilian Cerrado using PS monthly mosaics; and (ii) it presents a framework to select the best spectral and textural attributes to identify areas with double cropping based on the machine learning Random Forest classifier. Since there are no other PS monthly mosaics-based studies on mapping double cropping systems in the Cerrado biome, this work is intended to serve as a benchmark to help generate consistent and accurate LULC maps of the Brazilian Cerrado that include, for the first time, double cropping in their legends.

## 2. Materials and Methods

### 2.1. Study Area

The study area corresponds to the municipality of Goiatuba, covering an area of approximately 247,000 ha in the southern part of Goiás State, Brazil (latitude: $-18.00°$; longitude: $-49.60°$) (Figure 1). Goiatuba is becoming one of the most important municipalities of the Cerrado in terms of agricultural production in the south/southwest part of Goiás State. According to MapBiomas [6], 45% of this municipality was covered by annual crops in 2021, followed by sugarcane (17%) and cultivated pastures (9%). Nowadays, annual crops are replacing degraded pastures while sugarcane is advancing in croplands. For example, in this municipality, areas occupied by cultivated pastures decreased from 45,000 ha in 1985 to 23,000 ha in 2021 while harvested areas of soybean increased from 42,500 ha in 1985 to 76,000 ha in 2021, respectively [30,31].

The climate is tropical, with hot and humid summers and dry winters—Aw in the Köppen climate classification system [32]. According to the rain gauge station located in the neighboring municipality of Itumbiara (station code: 83523; latitude: $-18.41°$; longitude: $-49.19°$) [33], average annual precipitation is 1394 mm, concentrated between November (monthly average of 199 mm) and March (monthly average of 204 mm). The

topography is dominantly flat (47% of the municipality, typical slope of 0–3%), with surrounding depressed landforms (41%).

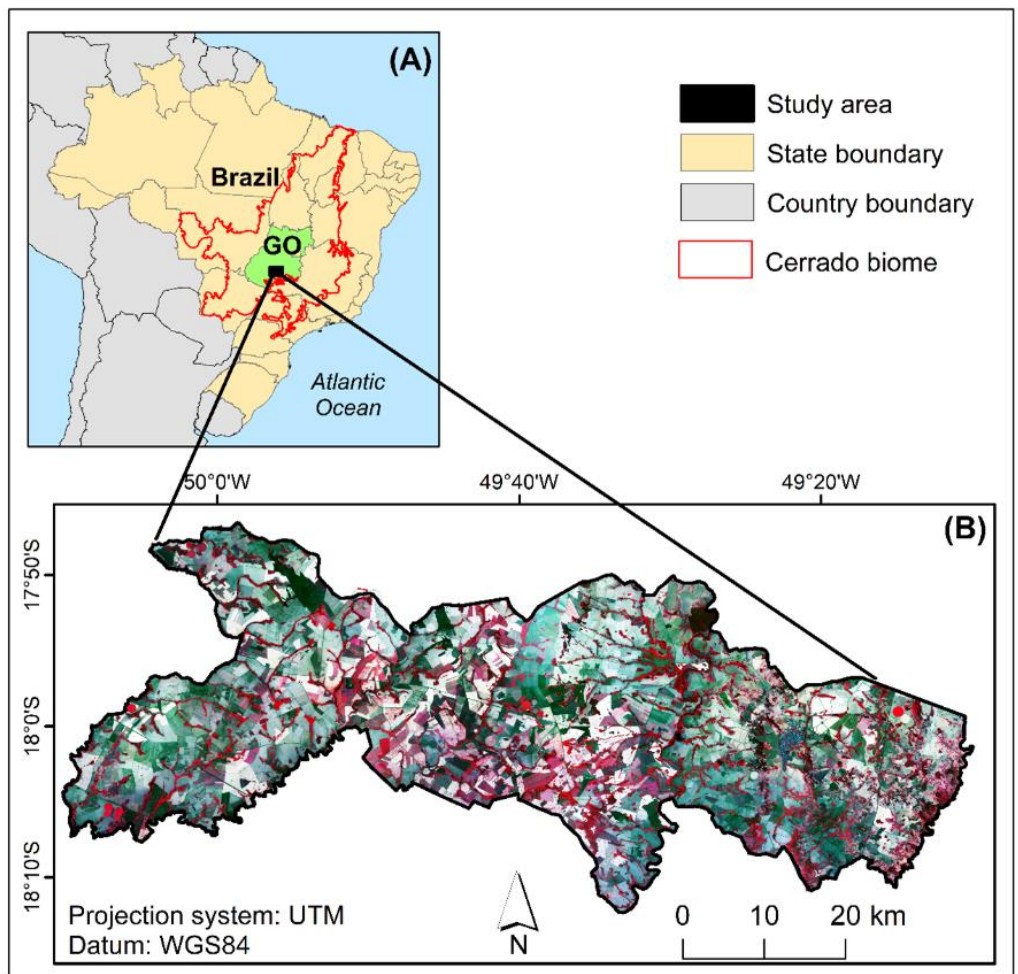

**Figure 1.** Location of the study area, municipality of Goiatuba in Goiás State, Brazil (**A**) and the RGB color composite of PlanetScope of bands 4 (near-infrared), 3 (red), and 2 (green) from October 2021 over the study area (**B**). GO = Goiás State.

Reddish, deep, and well-developed Ferralsols, according to the World Reference Base for Soil Resources (WRS) proposed by the Food and Agriculture Organization (FAO) [1] (Latossolo Vermelho in the Brazilian System of Soil Classification or Oxisols in the U.S. Soil Taxonomy), are the dominant type of soil in this municipality [34], which is highly weathered, deep (>2 m), well-drained, and with low levels of fertility and high contents of Si and Al toxicity [35]. This soil type is representative of the Cerrado biome where the two most common soil types are Ferralsols (44%) and Arenosols (21%). Figure 2 shows the ternary diagram of 153 soil texture analyses from this municipality. The soil samples were obtained at the 0–5 cm depth during the field campaign conducted in October 2022. The average percentages of sand, silt, and clay were 33%, 23%, and 44%, respectively, which is classified as fine texture in the FAO´s classification or loam in the U.S. Soil Taxonomy, that is, soils having roughly balanced proportions of sand, silt, and clay contents.

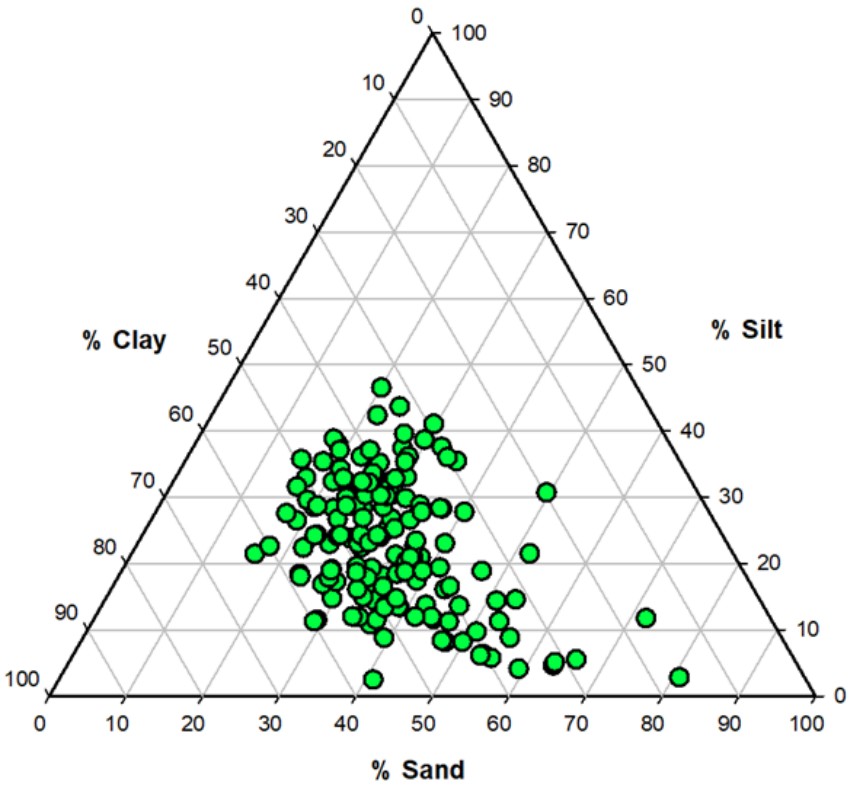

**Figure 2.** Ternary diagram of the soil texture from the municipality of Goiatuba, Goiás State, Brazil. Soil samples were obtained at a soil depth of 0–5 cm in October 2021.

*2.2. Methods*

Figure 3 shows the main steps of our methodological approach. We based our study on six monthly mosaics of PS satellite that were converted into spectral indices and textural attributes based on the Gray-Level Co-occurrence Matrix (GLCM) algorithm. The images were acquired during the 2021/2022 crop growing season. A combination of most relevant multispectral bands, spectral indices, and textural attributes was used as input parameters for the RF classifier [36]. The reference samples for training and validation of RF classification were obtained during two field campaigns conducted in October 2021 and April 2022.

We relied on the PS mosaics released by the NCIFI and made available for download in the Google Earth Engine cloud computing platform [37]. The mosaics are prepared primarily to combat deforestation and forest degradation and cover an area limited between 30° north latitude to 30° south latitude that corresponds to the world´s tropical region [38]. They are composed by the blue (0.455–0.515 µm), green (0.500–0.590 µm), red (0.590–0.670 µm), and NIR (0.780–0.860 µm) spectral bands and a spatial resolution of 4.77 m. Two mosaics per year were produced between the period of December 2015 to August 2020, becoming monthly since then.

In this study, we selected six normalized analytic mosaics from October 2021 to March 2022, which correspond to the crop growing season for most of the crop plantations in the study area. These mosaics are based on the PS surface reflectance data that are atmospherically corrected and normalized to reduce scene-to-scene variability [38]. The seamlines are also removed to minimize scene boundaries. According to Pandey et al. [38], PS normalized surface reflectance products work well whenever spatial consistency is prioritized and/or training data for machine learning algorithms are considered.

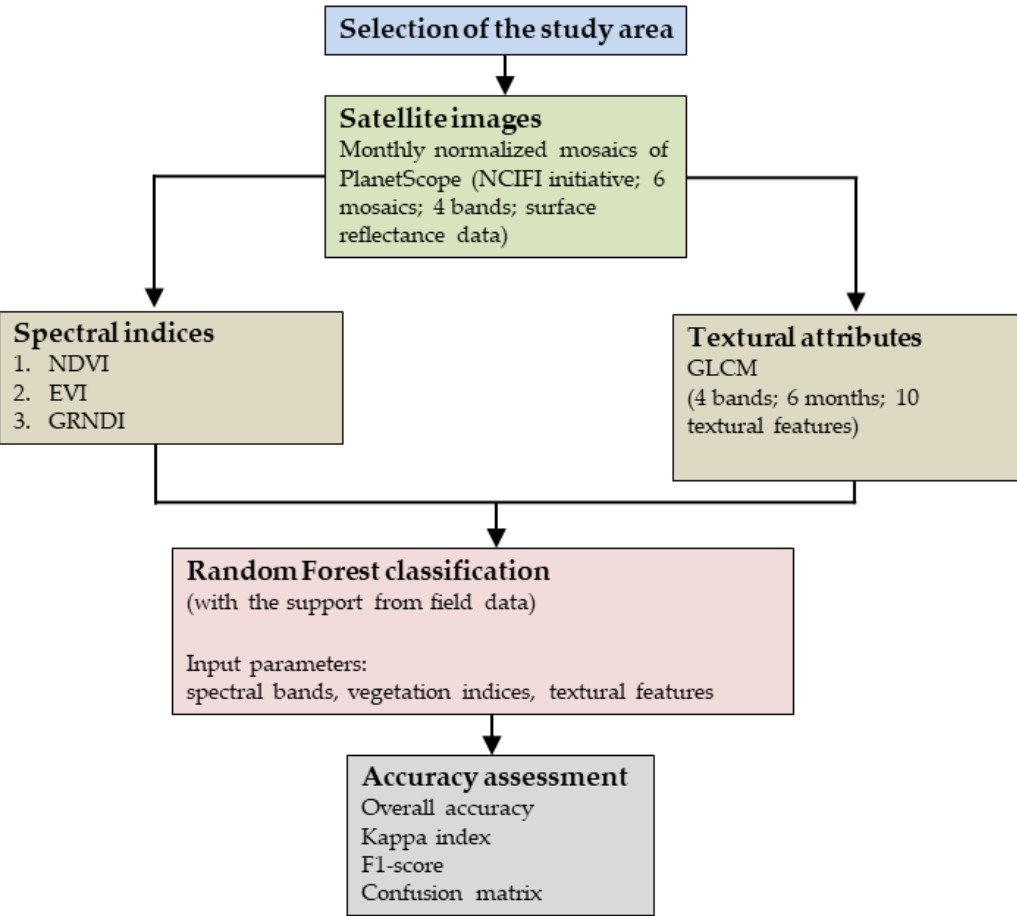

**Figure 3.** Flowchart showing the main steps of the methodological approach considered in this study. NCIFI = Norway´s International Climate and Forests Initiative; NDVI = Normalized Difference Vegetation Index; EVI = Enhanced Vegetation Index; GRNDI = Green–Red Normalized Difference Index; and GLCM = Gray-Level Co-occurrence Matrix.

For illustration purposes, Figure 4 shows three RGB color composites of PS mosaics of the study area (bands 3, 4, and 2, respectively) from October 2021, January 2022, and March 2022. In the October composite, greenish areas are mostly occupied by either evergreen native vegetation, sugarcane, or irrigated annual crops, while reddish areas correspond mainly to cultivated pastures and rainfed annual crops covered with crop residues. In January, there is a predominance of greenish color which corresponds to the peak of wet season in the study area, when most of the annual crops are also in the peak of crop growing season. In March (end of wet season), the areas with dominant greenish color are reduced in relation to the previous image since not all areas of second cropping are in the peak of growing season. The planting season for the second crop in this municipality usually occurs at the beginning of February (unpublished information provided by local producers).

We conducted a field survey in October 2021 to characterize the most representative LULC classes in the study area. We visited 201 sampling points that were previously defined based on visual interpretation of a Landsat 8 RGB color composite of bands 4, 5, and 6 acquired on 27 August 2021. We tried to include all representative LULC classes, both natural Cerrado formations (grasslands, shrublands, and forestlands) and land use classes (cultivated pastures, annual crops, semi-perennial crops, reforestation, and bare soils). These are the representative LULC classes of the municipality, according to the annual LULC maps produced by MapBiomas [6] for the entire country at the 30 m pixel size.

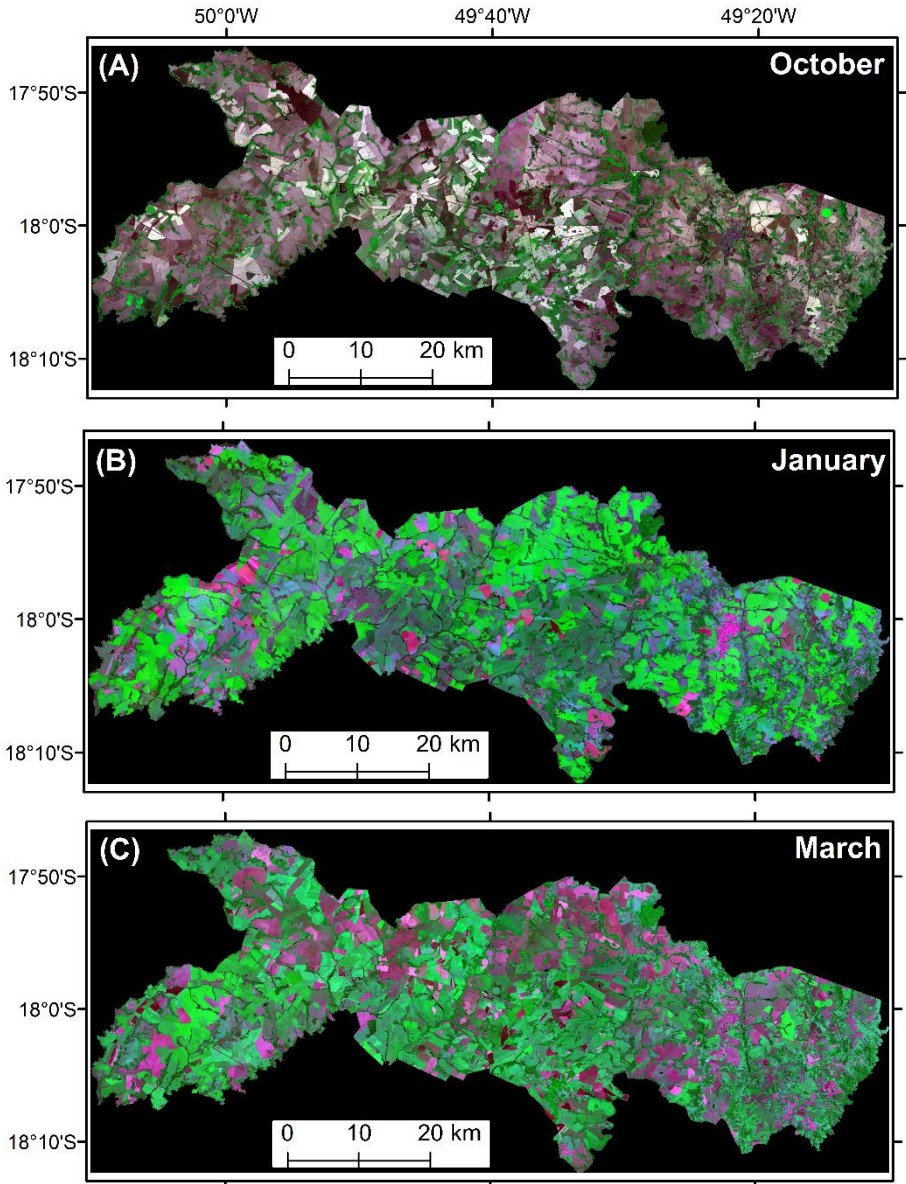

**Figure 4.** Monthly mosaics of PlanetScope RGB color composites of bands 3 (red), 4 (near-infrared), and 2 (green) from the municipality of Goiatuba, Goiás State, Brazil from October 2021 (**A**), January 2022 (**B**), and March 2022 (**C**).

For each site, we recorded its coordinates and LULC class and obtained panoramic pictures using an RGB digital camera with a resolution of 7.1 MPixel. Field data were collected using the AgroTag platform, developed by Embrapa Meio Ambiente, Jaguariúna, SP, Brazil, for recording georeferenced field data. AgroTag has an integrated WebGIS interface that allows access and analysis of data registered in this platform [39]. In the second field trip, conducted in April 2022, we revisited all points classified as annual crops and bare soils in October 2021 to estimate the level of adoption of double cropping by farmers as well as to record the type of second crop adopted by farmers. In the Goiatuba municipality, the most common second crops are maize and sorghum. Field data collected in this study are available upon request to the corresponding author.

Besides the four spectral bands of PS images converted into normalized surface reflectance, we included multitemporal information derived from vegetation indices and textural metrics as input parameters for the RF classification. We calculated three vegetation indices (VIs): Normalized Difference Vegetation Index (NDVI) [40] (Equation (1)), Enhanced

Vegetation Index (EVI) [41] (Equation (2)), and Green–Red Normalized Difference Index (GRNDI) [42] (Equation (3)).

$$NDVI = \frac{\rho_{NIR} - \rho_{Red}}{\rho_{NIR} + \rho_{Red}} \tag{1}$$

$$EVI = 2.5 \times \frac{\rho_{NIR} - \rho_{Red}}{\rho_{NIR} + 6 \times \rho_{Red} - 7.5 \times \rho_{Blue} + 1} \tag{2}$$

$$GRNDI = \frac{\rho_{Green} - \rho_{Red}}{\rho_{Green} + \rho_{Red}} \tag{3}$$

where $\rho$ is the surface reflectance in the blue, green, red, and NIR spectral bands.

These VIs take into consideration different combinations of PS spectral bands positioned in the visible and NIR wavelengths. NDVI is the most popular VI in the literature and depicts differences in photosynthetic activities in the plant canopies while EVI was developed to reduce NDVI-related constraints of signal saturation over dense vegetation cover as well as to minimize its soil and atmosphere effects [12,41]. GRNDI may detect seasonal changes in old/new foliage associated with seasonal changes of native vegetation [43] or crop growing processes.

We also derived textural features from the four spectral bands of the PS images based on the GLCM [44]. These attributes have been used by several authors for LULC classification in different landscapes and data sets [45,46]. The GLCM requires the setting of four parameters by users: Kernel´s window size, spectral band, level of quantization, and direction. Small windows (e.g., 3 × 3) may amplify noise while large windows (e.g., 31 × 31) may over-smooth texture [47]. However, other studies reported improvements in the crop classification when textural features derived from larger Kernel sizes were used [48].

Based on preliminary tests involving a randomly selected smaller area in the municipality of Goiatuba, we selected the following parameters: window size of 7 × 7, all four spectral bands, level of quantization of 64 bits, and azimuth direction of 45°. From this matrix, it is possible to extract 10 textural features from, for example, the open-source Sentinel Application Platform (SNAP) image processing software developed by the European Space Agency (ESA). However, some of these features may present redundant spatial context information. The most important textural features were selected based on the ranking of mean decrease accuracies [36] of the following metrics: contrast, dissimilarity, homogeneity, angular second moment, energy, entropy, mean, variance, correlation, and maximum probability. The equations and the definitions of each feature can be found in Iqbal et al. [46].

All attributes summarized in Table 1 were used as the input parameters for non-parametric, machine learning RF classification, with a 60%/40% training and test subset ratio. RF was proposed by Breiman [36] to improve the accuracy of image classification and regression trees through several combinations of random subsets of trees. Each tree contributes with one vote, and the final classification is produced considering the votes from all forest trees. We used the default value for the number of variables in the random subset at each node (mtry). The number of trees (ntree) was set as 300, which is the square root of the number of variables. Preliminary tests with 100, 300, and 500 trees showed that changing from 100 to 300 trees resulted in small improvements in the classification results. The ten most relevant combinations of spectral bands, spectral indices, and textural features were selected for the RF classification.

Using the validation dataset, we obtained the overall classification accuracy, Kappa index, F1-score, and omission and commission errors derived from a confusion matrix with a level of significance of 0.05. The overall accuracy and Kappa index provide an overview of the classification performance, while the omission and commission errors and the F1-score provide a by-class analysis. The overall accuracy refers to the number of correctly classified samples divided by the total number of samples, while Kappa index measures the degree of agreement of an image classification, and its values typically range from 0 (no agreement) to 1 (complete agreement). The main diagonal of the matrix indicates the correct classifications, while the off-diagonal elements indicate omission and commission

errors [49]. F1-score is derived from the precision (a relation between true positives and the total number of true positives and false positives) and recall (a relation between true positives and total number of true positives and false negatives) values and is considered as the harmonic mean of these two values.

**Table 1.** Description of input parameters derived from six PlanetScope monthly mosaics for Random Forest classification, seeking the discrimination of the double cropping plantations and other land use and land cover classes in the 2021–2022 crop growing season in the municipality of Goiatuba, Goiás State.

| Parameters | Description |
|---|---|
| Spatial resolution | 4.77 m |
| Processing level | Atmospherically corrected, normalized, analytic mosaics |
| Spectral bands | Blue (0.455–0.515 μm), green (0.500–0.590 μm), red (0.590–0.670 μm), and near-infrared (0.780–0.860 μm) |
| Vegetation indices | NDVI, EVI, GRNDI |
| Textural features | Contrast, dissimilarity, homogeneity, angular second moment, energy, entropy, mean, variance, correlation, and maximum probability |
| Monthly mosaics | October 2021 to March 2022 |

In addition to the classification performance estimators, a confidence map was also produced for the municipality of Goiatuba. This map indicates, on a per-pixel basis, the percentage of votes for the majority class (i.e., the class chosen for the pixel) out of the total votes from the RF decision trees [50]. High confidences indicate that the class is more likely to be correct as there was a clearer majority of votes by the ensemble of trees.

## 3. Results

### 3.1. Field Data

Table 2 summarizes the number of LULC classes visited in the two field campaigns. A total of 61 bare soils in October 2021, which correspond to the areas prepared for planting in this month, were revisited in April 2022. We found that only four sites were not used for second crop plantation. Maize and sorghum were the most common second crops in the municipality. Another six sites were planted with sugarcane.

**Table 2.** Summary of number of sites per land use and land cover (LULC) class visited during the field campaigns conducted in October 2021 and April 2022.

| LULC Class | Number of Sites Visited in October 2021 | LULC Class | Number of Sites Visited in April 2022 |
|---|---|---|---|
| Forestland | 36 | Forestland | – |
| Shrubland | 10 | Shrubland | – |
| Pastureland | 46 | Pastureland | – |
| Sugarcane | 45 | Sugarcane | – |
| Reforestation | 2 | Reforestation | – |
| Cotton | 1 | Cotton | – |
| Bare soil | 61 | Maize | 35 |
| | | Sorghum | 14 |
| | | Sugarcane | 6 |
| | | Crop residue | 4 |
| | | Crotalaria | 2 |
| TOTAL | 201 | | 61 |

### 3.2. Spectral Signatures and Indices

Figure 5 shows the spectral signatures of forestlands, cultivated pastures, single cropping, and double cropping found in the municipality of Goiatuba over the 2021/2022 crop growing season. The spectral signature of forestlands was derived based on averaging 36 sampling points that were visited during the field campaign of October 2021. A varying number of pixels was extracted from each sampling point, ranging from 418 to 555 pixels, with a total of 4300 pixels. In the same way, the spectral signature of cultivated pastures was obtained based on 46 sampling points (range: 254–646 pixels; total: 22,496 pixels). The spectral signature of double cropping was obtained considering the 49 field sampling points recorded as bare soil in October 2021 and as maize or sorghum in April 2022 (range: 402–604 pixels; total: 21,891 pixels). Finally, the spectral signature of single cropping was obtained through six sampling points recorded as bare soil in October 2021 and sugarcane in April 2022 (range: 418–552 pixels; total: 4300 pixels). The typical spectral signature of sugarcane (older than one year) is not shown in this paper because of varying plant growing conditions found in the study area.

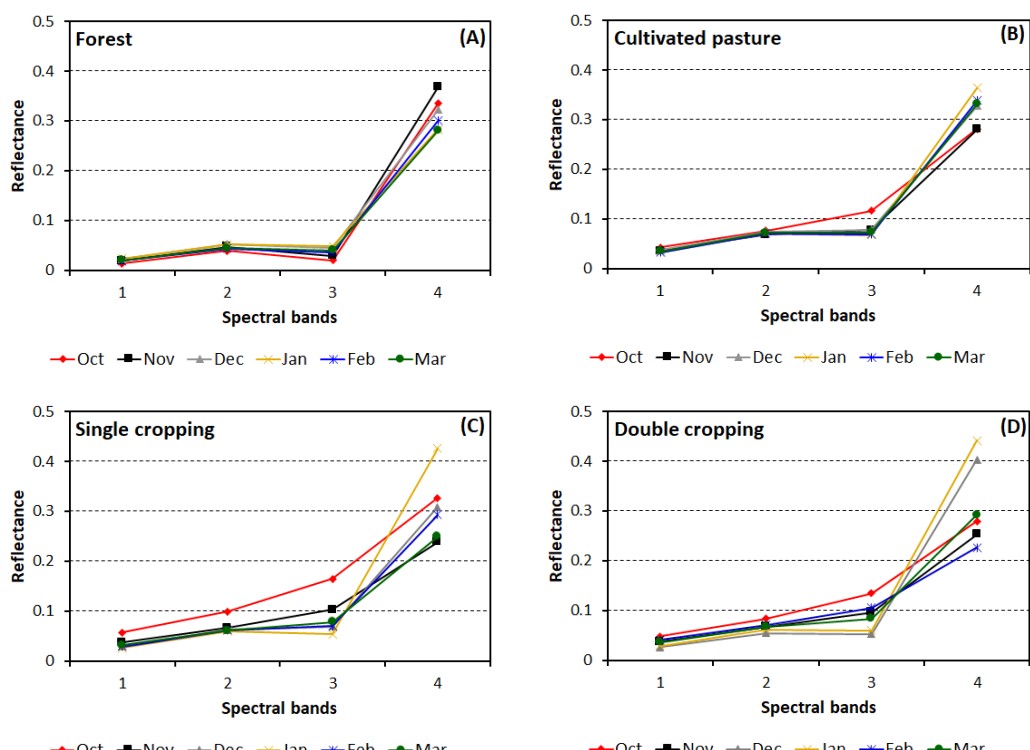

**Figure 5.** Multitemporal surface reflectance of forestlands (**A**), cultivated pasturelands (**B**), single croplands (**C**), and double croplands (**D**) from the municipality of Goiatuba, Goiás State, Brazil (2020/2021 crop growing season). The bands 1, 2, 3, and 4 represent the spectral bands of PlanetScope satellite in the blue, green, red, and near-infrared ranges, respectively.

The forestlands presented spectral signatures that are typical of those from green leaves, that is, low reflectance in the red band due to the absorption of electromagnetic radiation related to the photosynthetic activities and high reflectance in the NIR band due to the high reflection of solar incident radiation controlled by the internal structure of leaves during the entire time series. Similar signatures were also found for shrubland and reforestation (not show in this figure).

In October, cultivated pastures did not show strong absorption in the red band or strong reflection in the NIR band, indicating that the pastures were mostly dry. The main difference between the temporal signatures of areas with single cropping and double cropping is in the red and NIR spectral bands from December and January. Most of the double cropping areas are in the peak of growing season in December and early January, while the harvesting

season is in early February. The single crop spectral signature from December does not show the typical reflectance of green vegetation because most of these areas were recently planted with sugarcane, which presents a much longer crop growing season.

The most stable EVI values during the 2021/2022 crop growing season were found in the forestlands (typical range: 0.40–0.65) (Figure 6). Although we found some sites with deciduous forest, most of the forestlands in the study area were composed by gallery forests and semideciduous forests. As expected, the cultivated pastures were driest in October (lowest EVI values, typically around 0.25) and greenest in January, the peak of rainy season in the study area. The most impressive aspect is the difference in EVI values between single cropping and double cropping from February. EVI from double cropping showed much lower EVI values (average: 0.19) than those from single cropping (average: 0.37). The variance of EVI values for single cropping was also much higher than that from double cropping for this period.

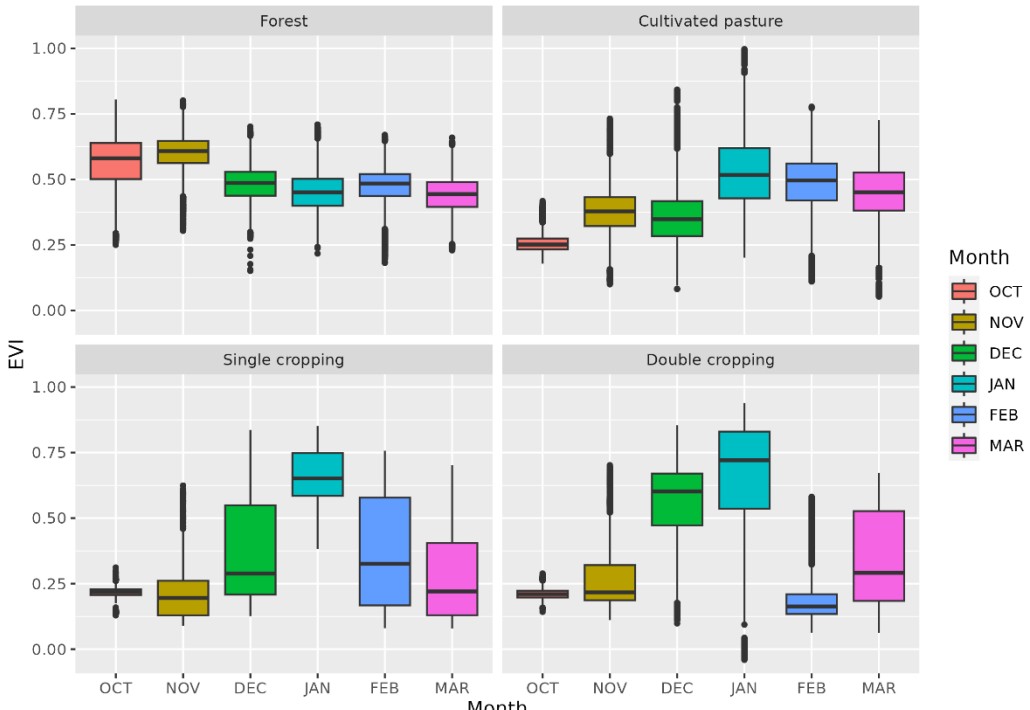

**Figure 6.** Multitemporal enhanced vegetation index (EVI) of forestlands, cultivated pasturelands, single croplands, and double croplands from the municipality of Goiatuba, Goiás State, Brazil (2020/2021 crop growing season).

### 3.3. Textural Features

We found that some GLCM textural features were highly correlated regardless of spectral band or month of the year. This is the case, for example, between asymmetry and maximum probability (r = 0.969) or between mean and variance (r = 0.973) derived from the mosaic obtained in October 2021 and band 4 (Table 3). In this figure, there are 13 combinations with coefficients of correlation higher than 0.80, which agrees with the statement made by ESA [51] that some textural features produced by the GLCM can present some redundancy. Therefore, further analysis of reduction of data dimensionality for optimizing processing was addressed (see Table 3 below).

**Table 3.** Coefficients of correlation of 10 GLCM textural features derived from band 4 of PlanetScope mosaic obtained in October 2021. All positive and negative correlations higher than 0.80 are highlighted in red color. ASM = angular second moment; CON = contrast; COR = correlation; DIS = dissimilarity; ENE = energy; ENT = entropy; HOM = homogeneity; MAX = maximum value; MEAN = mean value; VAR = variance.

|      | ASM | CON | COR | DIS | ENE | ENT | HOM | MAX | MEAN | VAR |
|------|-----|-----|-----|-----|-----|-----|-----|-----|------|-----|
| ASM  | 1   |     |     |     |     |     |     |     |      |     |
| CON  | −0.343 | 1 |     |     |     |     |     |     |      |     |
| COR  | 0.067 | −0.381 | 1 |    |     |     |     |     |      |     |
| DIS  | −0.561 | 0.924 | −0.358 | 1 |  |     |     |     |      |     |
| ENE  | 0.957 | −0.477 | 0.178 | −0.719 | 1 |   |     |     |      |     |
| ENT  | −0.772 | 0.641 | −0.234 | 0.862 | −0.912 | 1 |  |     |      |     |
| HOM  | 0.822 | −0.636 | 0.180 | −0.869 | 0.937 | −0.966 | 1 |  |      |     |
| MAX  | 0.969 | −0.400 | 0.084 | −0.628 | 0.975 | −0.834 | 0.880 | 1 |  |     |
| MEAN | 0.025 | 0.049 | 0.423 | 0.059 | 0.149 | −0.086 | −0.038 | 0.006 | 1 |  |
| VAR  | 0.149 | 0.006 | 0.359 | −0.012 | 0.266 | −0.189 | 0.061 | 0.123 | 0.973 | 1 |

Table 4 shows the statistical results of RF classification involving all spectral bands and vegetation indices and 20 out of 240 most important combinations involving textural features (10 textural features × 6 months × 4 bands). As the option with the 10 combinations presented the highest accuracies (overall accuracy = 90.91%; Kappa index = 0.8784), we selected this option for producing the final RF classification map of the municipality of Goiatuba. We chose to generate the final map using the Orfeo Toolbox (OTB) plug-in in the QGIS 3.10 software because of the confidence map, an output that represents the level of confidence in the classification of each pixel considering the proportion of votes for the majority class [51].

**Table 4.** Statistical results of Random Forest classification involving 4 spectral bands, 3 vegetation indices, and 20 out of 240 possible combinations involving Gray-Level Co-occurrence Matrix (GLCM) textural features (10 GLCM features × 6 months × 4 bands).

| Model | Attributes | Overall Accuracy (%) | Kappa Index | $p$-Value | mTry | nTree |
|-------|-----------|----------------------|-------------|-----------|------|-------|
| 1  | 4 Bands + 3 VIs + 1 GLCM   | 88.31 | 0.8438 | $<2.2 \times 10^{-16}$ | 6 | 300 |
| 2  | 4 Bands + 3 VIs + 2 GLCMs  | 88.31 | 0.8438 | $<2.2 \times 10^{-16}$ | 7 | 300 |
| 3  | 4 Bands + 3 VIs + 3 GLCMs  | 89.61 | 0.8611 | $<2.2 \times 10^{-16}$ | 7 | 300 |
| 4  | 4 Bands + 3 VIs + 4 GLCMs  | 89.61 | 0.8611 | $<2.2 \times 10^{-16}$ | 7 | 300 |
| 5  | 4 Bands + 3 VIs + 5 GLCMs  | 89.61 | 0.8611 | $<2.2 \times 10^{-16}$ | 7 | 300 |
| 6  | 4 Bands + 3 VIs + 6 GLCMs  | 89.61 | 0.8611 | $<2.2 \times 10^{-16}$ | 7 | 300 |
| 7  | 4 Bands + 3 VIs + 7 GLCMs  | 89.61 | 0.8611 | $<2.2 \times 10^{-16}$ | 7 | 300 |
| 8  | 4 Bands + 3 VIs + 8 GLCMs  | 89.61 | 0.8611 | $<2.2 \times 10^{-16}$ | 7 | 300 |
| 9  | 4 Bands + 3 VIs + 9 GLCMs  | 90.91 | 0.8780 | $<2.2 \times 10^{-16}$ | 7 | 300 |
| 10 | 4 Bands + 3 VIs + 10 GLCMs | 90.91 | 0.8784 | $<2.2 \times 10^{-16}$ | 7 | 300 |
| 11 | 4 Bands + 3 VIs + 11 GLCMs | 88.31 | 0.8437 | $<2.2 \times 10^{-16}$ | 7 | 300 |
| 12 | 4 Bands + 3 VIs + 12 GLCMs | 88.31 | 0.8436 | $<2.2 \times 10^{-16}$ | 7 | 300 |
| 13 | 4 Bands + 3 VIs + 13 GLCMs | 88.31 | 0.8436 | $<2.2 \times 10^{-16}$ | 7 | 300 |
| 14 | 4 Bands + 3 VIs + 14 GLCMs | 89.61 | 0.8611 | $<2.2 \times 10^{-16}$ | 7 | 300 |
| 15 | 4 Bands + 3 VIs + 15 GLCMs | 89.61 | 0.8610 | $<2.2 \times 10^{-16}$ | 7 | 300 |
| 16 | 4 Bands + 3 VIs + 16 GLCMs | 89.61 | 0.8610 | $<2.2 \times 10^{-16}$ | 8 | 300 |
| 17 | 4 Bands + 3 VIs + 17 GLCMs | 88.31 | 0.8436 | $<2.2 \times 10^{-16}$ | 8 | 300 |
| 18 | 4 Bands + 3 VIs + 18 GLCMs | 88.31 | 0.8435 | $<2.2 \times 10^{-16}$ | 8 | 300 |
| 19 | 4 Bands + 3 VIs + 19 GLCMs | 88.31 | 0.8437 | $<2.2 \times 10^{-16}$ | 8 | 300 |
| 20 | 4 Bands + 3 VIs + 20 GLCMs | 88.31 | 0.8437 | $<2.2 \times 10^{-16}$ | 8 | 300 |

Figure 7 shows the 10 most relevant input parameters for the RF classification based on the mean decrease accuracy criterion. This figure also shows the importance of PS mosaics from February, a month in which the seven top-ranked input parameters are involved. In terms of GLCM attributes, the variance, mean, and correlation were the most relevant parameters.

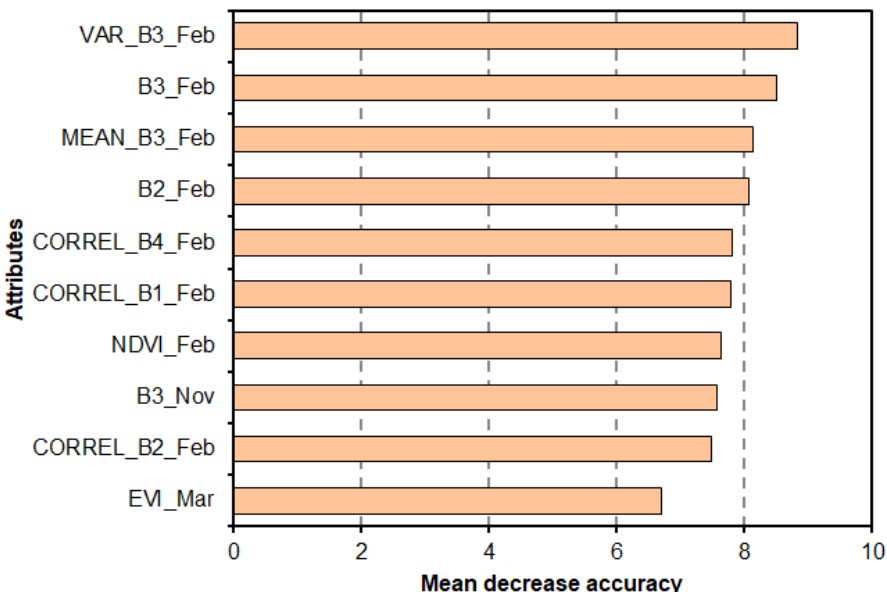

**Figure 7.** Rank of 10 most important attributes for Random Forest classification. See Figure 7 for identification of textural attribute short names. NDVI = normalized difference vegetation index; GRNDI = green–red normalized difference index; EVI = enhanced vegetation index.

### 3.4. Random Forest Classification

Figure 8 shows the final map of RF classification, after smoothing the classification result by using a 4 × 4 window size filter. The dominant LULC class was the double cropped area (37%), followed by cultivated pasture (22%) and sugarcane (21%). The majority of the municipality (80%) is dominated by crop, cattle meat, sugar, and biofuel production, while native vegetation occupies only 18% of the municipality, contrasting with the average portrait of the Cerrado biome, which is roughly occupied by 50% of agriculture and 50% of native vegetation.

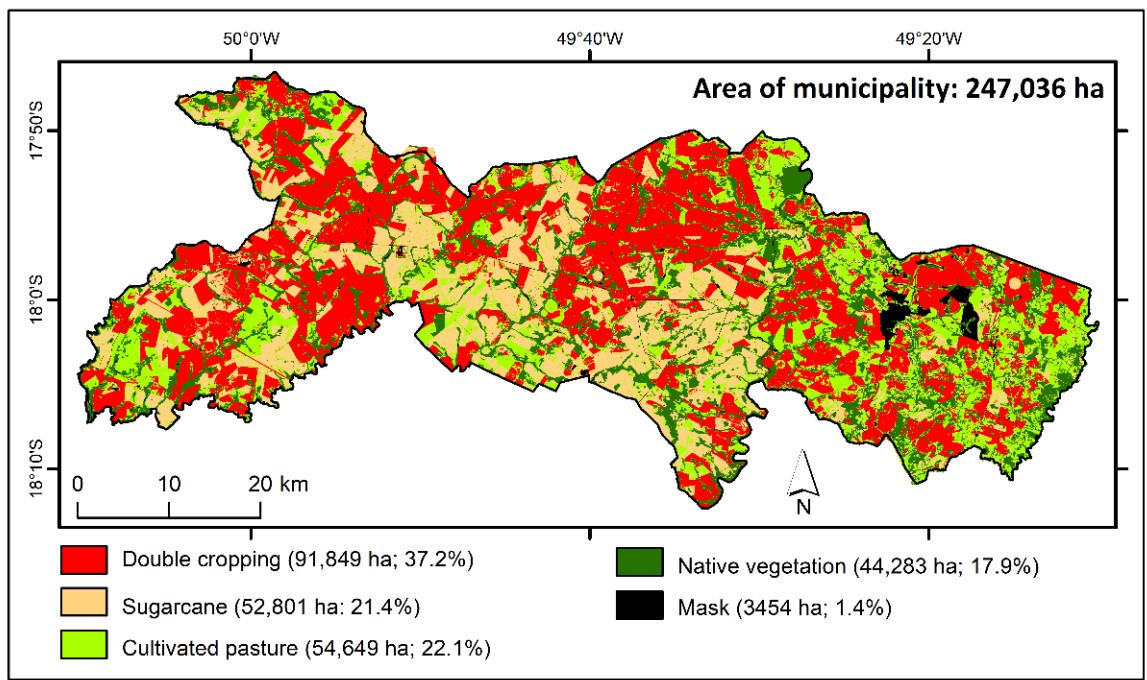

**Figure 8.** Land use and land cover (LULC) map of the municipality of Goiatuba, Goiás State, Brazil, showing the dominant presence of double cropping areas.

*3.5. Accuracy Assessment*

The overall accuracy and the Kappa index of the final map were 90.91% and 0.8784, respectively (Table 5). Class-specific omission errors ranged from 0 for native vegetation and cultivated pasture to 20% for sugarcane, while commission errors ranged from 0 for native vegetation to 26% for cultivated pasture. For double cropping, the omission and commission errors were 13% and 5%, respectively, while the F1-score was 0.91. In other words, from a total of 24 field validation sites of double cropping, one site was misclassified as single crop and two sites as cultivated pasture (omission error). On the other hand, one single crop field was misclassified as double crop (commission error). There was no confusion to discriminate natural vegetation, while single crop presented the highest omission error (20%). In the same way, cultivated pastures presented the highest commission error (26.3%).

**Table 5.** Confusion matrix for Random Forest classification based on the PS mosaics for municipality of Goiatuba, Goiás State, with 95% of confidence interval. DC = double cropping; SC = sugarcane; CP = cultivated pasture; NV = native vegetation; O.E. = omission error; C.E. = commission error.

| | | **RF Classification** | | | | | | | | |
|---|---|---|---|---|---|---|---|---|---|---|
| | | DC | SC | CP | NV | Total | O.E. (%) | C.E. (%) | Precision | Recall | F-score |
| Field Data | DC | 21 | 1 | 2 | 0 | 24 | 13.0 | 4.5 | 0.95 | 0.88 | 0.91 |
| | SC | 1 | 16 | 3 | 0 | 20 | 20.0 | 5.9 | 0.94 | 0.80 | 0.86 |
| | CP | 0 | 0 | 14 | 0 | 14 | 0 | 26.3 | 0.74 | 1 | 0.85 |
| | NV | 0 | 0 | 0 | 19 | 19 | 0 | 0 | 1 | 1 | 1 |
| | Total | 22 | 17 | 19 | 19 | 77 | | | | | |
| | Overall accuracy (%) | 90.91 | | | | | | | | | |
| | Kappa index | 0.8784 | | | | | | | | | |

Figure 9 shows the classification confidence map of the LULC classification result for the municipality of Goiatuba. The majority of the municipality presented overall high confidence (81.0% ± 18.8%) (Figure 10). The classification confidence was high for double cropping (87.6% ± 16.1%) and native vegetation (88.0% ± 16.7%) and relatively low for sugarcane (73.3% ± 18.3%) and cultivated pasture (75.5% ± 17.3%).

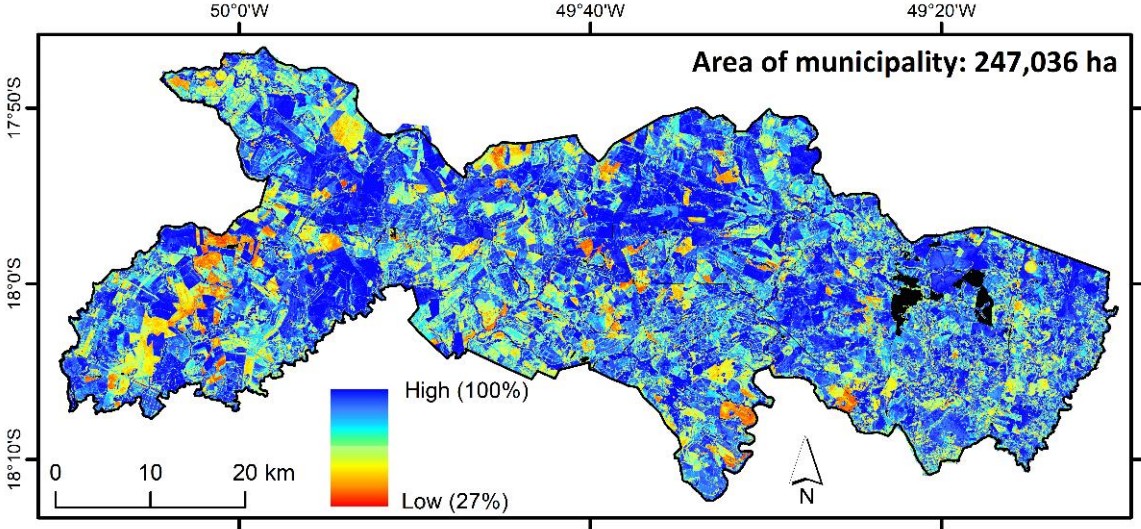

**Figure 9.** Confidence map for the 2021/2022 crop growing season of the municipality of Goiatuba, Goiás State, Brazil, based on percentage of majority votes from the Random Forest (RF) classifier. The label "High (100%)" in the legend means that all RF trees voted for the final class.

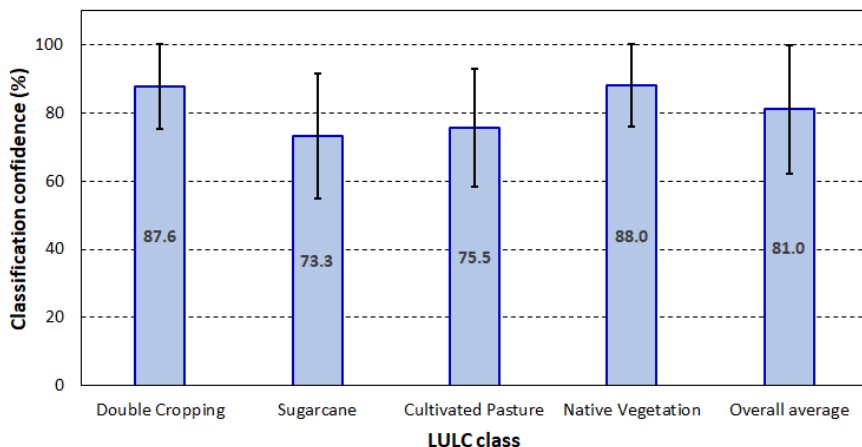

**Figure 10.** Percentage of classification confidence for the 2021/2022 crop growing season in the municipality of Goiatuba, Goiás State, Brazil, discriminated by land use and land cover (LULC) classes considered in the Random Forest classification.

## 4. Discussion

In the municipality of Goiatuba, Goiás State, Brazil, we found 1330 rural properties registered in the Brazilian Rural Environmental Registering system (CAR—Cadastro Ambiental Rural) on 8 December 2022. The average size of farms in this municipality was 171 ha [52], much smaller than the average size of farms located in the Brazilian Amazon (e.g., 1737 ha; 615 farms with an average size of 1737 ha in the municipality of Sapezal, Mato Grosso State). This means that the recent investigations, such as the one conducted by Chaves et al. [19] in the municipality of Sapezal to distinguish single cropping from double cropping based on the MODIS time series data processed through the TWDTW strategy, may not work properly. In other words, the MODIS data processed by the TWDTW work well in large farms such as those found in agricultural frontiers located in the transition zone between Cerrado and Amazonia biomes, which are the cases of the municipalities of Lucas do Rio Verde, Sapezal, and Sorriso, among others. For smaller farms, the much better spatial resolution of PS monthly mosaics (4.77 m) offers a more promising possibility of discriminating single cropping from double cropping with high accuracy.

In general, most of the agricultural activities in the Cerrado biome using remote sensing data have been monitored based on optical satellite images with a broad range of spatial resolution (10–250 m), often converted into spectral indices or phenological metrics [5,53]. To the best of our knowledge, only the study conducted by Vizzari [53] has used GLCM-related textural features derived from a six-month PS mosaic to produce an LULC map over the western Bahia State, Brazil. The image classification based on this object-based strategy showed higher performance (overall accuracy of 82%) than the classification based on pixel-based strategy (overall accuracy of 67%).

The GLCM-derived textural features have been more related to the radar remote sensing and to the very high spatial resolution images obtained by unmanned aerial vehicles (UAVs). There are a number of studies available in the literature using these attributes derived from synthetic aperture radar (SAR) data to identify clear-cut deforested areas, selective logged areas, and different stages of secondary vegetation. The UAV-derived optical images have been more frequently used in the agricultural applications. In theory, GLCM-related textural features derived from both the UAV platform and PS mosaics can provide relevant information for identifying different types of LULC classes in the Cerrado biome, since crop planting rows with varying distance, different levels of biomass, and different azimuth angles, among other aspects, produce distinct textural patterns not only in SAR images, but also in optical images.

In the list of our 10 most important input parameters for the RF classification, we found the following textural features: correlation, variance, and mean from the mosaic of February. The selection of most relevant textural features is both site-specific and sensor-specific, as

confirmed by other studies such as the one conducted by Sothe et al. [47], who reported contrast and dissimilarity as the most important textural attributes associated with the infrared spectral bands of Landsat 8 and RapidEye satellite images. The selection of most important textural features is crucial to reduce computational demand for running RF and other non-parametric machine learning or deep learning classifiers. Besides, as demonstrated in our study, increasing the number of input parameters for the RF classification does not assure increase in the accuracy of the classification results.

The RF classification showed that 82% of the municipality is covered by different types of land occupation, which is in disagreement with the Brazilian Forest Code, which determines a maximum of 80% of land occupation (see more details in the Introduction Section). Our results of RF classification present a relatively good agreement with those from the MapBiomas initiative [6] (Figure 11), except for cultivated pasture. PS mosaics underestimated annual crops by 8%. This difference was somewhat expected because of differences in the image acquisition modes, matching mapping legends, and satellite overpasses. According to IBGE [30], the total harvested area with annual crops in the municipality of Goiatuba was 176,032 ha in 2021, that is, 71% of the municipality, a percentage much higher than those estimated by PS (37%) or MapBiomas (45%). This difference is because this institution (Brazilian Institute of Geography and Statistics—IBGE) counts the harvested area twice whenever the double cropping system is adopted. For example, Goiatuba harvested 27,400 ha of corn in 2021, but 94% of this area corresponded to the second crop [54].

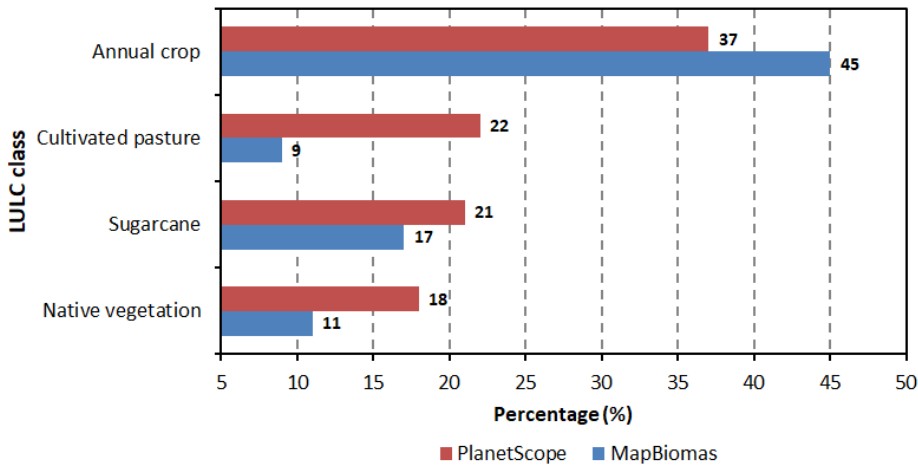

**Figure 11.** Comparison between the land use and land cover (LULC) data produced by the PlanetScope monthly mosaics and by the MapBiomas initiative for the municipality of Goiatuba, Goiás State, Brazil.

Some authors have used Bayesian-based dropouts during the calibration phase of the methodology so that the model uncertainty can be statistically estimated [55,56]. In practice, this method, known as Monte Carlo dropout, has become attractive since it deals well with large amounts of data [57]. We recommend testing this approach in order to investigate the radiometric consistency of PS monthly mosaics in terms of signal variations associated with varying image acquisition modes and multiple dates of individual satellites that are part of the PS constellation.

Regarding the limitations of this study, we can point out the differences in the crop type (mostly for two types—maize and sorghum) and main differences in the crop planting dates that can cause large radiometric variations in the PS monthly mosaics, leading to misclassifications. As pointed out above, the radiometric variations due to different image acquisition modes and different dates of individual PS satellites that are considered in the production of monthly mosaics are another source of classification errors, although these mosaics are processed for normalization to reduce scene-to-scene variability and for seamline removal to minimize scene boundaries [38].

## 5. Conclusions

Results of this study allowed elucidation of the following conclusions:

a.  The high spatial and temporal resolution of the PS constellation of over 200 nanosatellites allows the addition of "double cropping" into the legend of LULC maps of the Cerrado biome.
b.  The use of textural attributes derived from the GLCM as input parameters in the supervised machine learning classification procedures is highly recommended.
c.  The most relevant PS images for identifying double cropping farm management systems are the ones obtained in February.

The products and the models selected in this work can be used as a powerful public policy to monitor agricultural intensification in the Cerrado biome or in other biomes in an operational way. As the texture attributes showed good performance to identify double-cropped fields, these metrics derived from GLCM can be tested to identify crop-livestock-forest integration systems in Brazil, which have been increasingly adopted in the country. The methodological approach used in this study is a good candidate to be a benchmark to generate consistent and accurate LULC maps of the Brazilian Cerrado with the unprecedented presence of the double cropping in their legends.

**Author Contributions:** Conceptualization, E.E.S. and É.L.B.; methodology, E.E.S.; validation, E.E.S.; formal analysis, E.E.S.; investigation, E.E.S. and É.L.B.; resources, É.L.B. and E.E.S.; data curation, E.E.S. and T.C.P.; writing—original draft preparation, E.E.S.; writing—review and editing, E.E.S., É.L.B., I.D.S., D.d.C.V., L.E.V., T.C.P. and G.M.B.; visualization, T.C.P.; supervision, E.E.S. All authors have read and agreed to the published version of the manuscript.

**Funding:** This research was funded by the São Paulo Research Foundation (FAPESP), grant # 2019/26222-6 "Agricultural mapping in the Cerrado via combined use of multisensor images" (Édson Bolfe) and by the National Council for Scientific and Technological Development (CNPq)—grant # 406494/2018-5 "Analysis of possibility of mapping abandoned agricultural areas in the Cerrado through Google Earth Engine" (Edson Sano) and Research Productivity Fellowship of Edson Sano (grant # 303502/2019-3), Édson Bolfe (302706/2019-4), and Ieda Sanches (310042/2021-6). This study was also partially financed by the Coordenação de Aperfeiçoamento de Pessoal de Nível Superior (CAPES), Brazil, Finance Code 001.

**Informed Consent Statement:** Not applicable.

**Data Availability Statement:** The data presented in this study are available upon request to the corresponding author.

**Conflicts of Interest:** The authors declare no conflict of interest.

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
