# Peer review of "Estimating Double Cropping Plantations in the Brazilian Cerrado through PlanetScope Monthly Mosaics"

_land, doi:10.3390/land12030581_

Round 1

Reviewer 2 Report

About the submission with the title "Estimating Double Cropping Plantations in the Brazilian Cerrado through PlanetScope Monthly Mosaics" I have the following comments:

There are already several studies addressing these approaches. From the abstract and the introduction section is difficult to understand the gaps adressed by this research and the main contributions for the scientific literature. Maybe a better literature review may brings new lights about these questions.

I suggest to create new section for the results and discussions. Another question is the number of subsections. I suggest to merge some of them.

A benchmarking with the results obtained by other studies and a benchmarking with the approaches considered by other researches are missing. The study is interesting, but it seems that does not bring nothing of new. Please search some systematic reviews available about these topics on the literature and see which was already done and which is lacking to do.

The conclusions section should be prepared with policy recommendations, practical implications and future research.

Reviewer 3 Report

This manuscript focuses on the potentiality of PS monthly mosaics to detect double cropping systems and other land use and land cover (LULC) classes in the Brazilian Cerrado, is well-written and could make a positive impact on the field of sustainable land management. However, there are a few issues with the paper that need to be fixed in order to make it better overall.

Comments and suggestions:

1.       Although several research are currently accessible, reviews on the potential of PS's satellite data in land use categorization are lacking from the introduction. To make the literature review better, the author should include recently completed research that made use of PS monthly mosaic data.

2.       Table 2 shows that very few sites were visited for reforestation, cotton, sugarcane, fallow, and crotalaria. Why did the author only use a small number of samples for these land use types? Is this a good number of samples to prove the accuracy?

3.       Usually, in tropics, clouds and shadows reduce the effectiveness of optical sensors for observation. On the other hand, the Planet microsatellite constellation is made up of many individual satellites, each with its own unique sensor properties and therefore might have impacted the capacity to acquire consistent SR across a broad area. Besides, LULC classification uncertainty may also be caused by variations in satellite product versions, atmospheric and directional adjustments, and others. All of these aspects not only impact the capacity to monitor double cropping directly, but also create inter- and intra-class similarities and class differences, leading to subpar classification results. In order to investigate how these elements play a role in the final categorization, the author may use a Monte Carlo dropout-generated uncertainty mask to remove the pixels that were impacted. This is important to justify the higher accuracy of the result found in this study.

4.       The author perhaps did not remark any of the limitations of the research; nonetheless, in modeling and geospatial analysis that is based on satellite data, several limitations are interconnected. So, I would suggest pointing out some of the limitations at the end of the discussion section.

Reviewer 4 Report

This is a very competent manuscript.  It is well written and presented.  There is a clear statement of intent and a logical flow to the text with appropriate transitions between sections.  The figures and tables are informative and professional.  There is an extensive set of appropriate references in consistent and complete format.

The topic of mapping double cropping agriculture via remote sensing is also important given the issues of food production globally and the increase in double cropping.  The authors also clearly explained the challenges of cloud cover in their study area and thus the selection of data sources.  The authors are also to be congratulated on their field efforts which are very important to most remote sensing studies but not always consistently employed.  The study can serve as a model for other locations and applications.  The examination of different vegetation indexes as well as texture measures is also compelling and potentially useful to other scientists.

As in almost all manuscripts, there are some editorial suggestions as follows for consideration by the authors:

1.       Line 24 and elsewhere such as line 350, Table 4 etc.  Overall accuracy should be a %.

2.       It would be useful to discuss if double cropping only includes the same crop twice or possibly different crops.  It is somewhat confusing from the text.

3.       Line 106. (red), add serial comma.

4.       Line 122. (21%) . or ,?

5.       Line 168, from not form.

6.       Line 171. Straw or straws?  Not clear, perhaps crop residue.

7.       Line 313. Does for do.

8.       Line 333. Table not Figure.

9.       Line 485.  Not hundreds.

10.   Line 523.  Int. in italics.

As stated, this is an excellent manuscript and certainly appropriate for this journal.

Round 2

Reviewer 2 Report

The gaps adressed and the novelties are not yet clear. I suggest to clearly address these parts, because there are many studies about these topics and the question here is where this paper is different from the others.

Author Response

The gaps adressed and the novelties are not yet clear. I suggest to clearly address these parts, because there are many studies about these topics and the question here is where this paper is different from the others.

We revised the Introduction section substantially, adding new paragraphs in order to attend this suggestion properly. We hope this important flaw raised by the reviewer was addressed satisfactorily.

Reviewer 3 Report

The authors have made certain modifications in response to the reviewers' suggestions, which have enhanced the quality to a certain degree. But I would suggest again that the author rethink the accuracy assessment and uncertainty analysis, if at all possible. This could make the paper more interesting.

Author Response

The authors have made certain modifications in response to the reviewers' suggestions, which have enhanced the quality to a certain degree. But I would suggest again that the author rethink the accuracy assessment and uncertainty analysis, if at all possible. This could make the paper more interesting.

We thank you again for this comment by the reviewer. The last two paragraphs of the Methods section, which details the accuracy and uncertainty analyses were expanded as:

Using the validation dataset, we obtained the overall classification accuracy, Kappa index, F1-score, and omission and commission errors derived from confusion matrix with a level of significance of 0.05. The overall accuracy and Kappa index provide an overview of the classification performance, while the omission and commission errors and the F1-score provide a by-class analysis. The overall accuracy refers to the number of correctly classified samples divided by the total number of samples while Kappa index measures the degree of agreement of an image classification and its values typically ranges from 0 (no agreement) to 1 (complete agreement). The main diagonal of the matrix indicates the correct classifications while the off-diagonal elements indicates omission and commission errors [49]. F1-score is derived from the precision (a relation between true positives and the total number of true positives and false positives) and recall (a relation between true positives and total number of true positives and false negatives) values and is considered as the harmonic mean of these two values.

In addition to the classification performance estimators, a confidence map was also produced for the municipality of Goiatuba. This map indicates, on a per-pixel basis, the percentage of votes for the majority class (i.e., the class chosen for the pixel) out of the total votes from the RF decision trees [50]. High confidences indicate that the class is more likely to be correct as there was a clearer majority of votes by the ensemble of trees.

In the Results section, we created a new subsection named 3.5. Accuracy Assessment. In this subsection, we added the F1-score data in the revised version of the manuscript. A new figure showing the percentage of classification confidence per LULC class was also added.
